# Remote Sensing Estimation and Spatiotemporal Pattern Analysis of Terrestrial Net Ecosystem Productivity in China

**Liang Liang *** , **Di Geng, Juan Yan, Siyi Qiu, Yanyan Shi, Shuguo Wang, Lijuan Wang, Lianpeng Zhang and Jianrong Kang**

School of Geography, Geomatics and Planning, Jiangsu Normal University, Xuzhou 221116, China; gengdi@jsnu.edu.cn (D.G.); juanyan@jsnu.edu.cn (J.Y.); qiusiyi@jsnu.edu.cn (S.Q.); syyhealer@jsnu.edu.cn (Y.S.); swang@jsnu.edu.cn (S.W.); wanglj2013@jsnu.edu.cn (L.W.); lianpeng_rs@jsnu.edu.cn (L.Z.); kangjianrong@jsnu.edu.cn (J.K.)
**\*** Correspondence: liang_rs@jsnu.edu.cn; Tel.: +86-187-9623-7312

**Abstract:** Net ecosystem productivity (NEP) plays an important role in understanding ecosystem function and the global carbon cycle. In this paper, the key parameters of the Carnegie Ames Stanford Approach (CASA) model, maximum light use efficiency ($\varepsilon_{max}$), was optimized by using vegetation classification data. Then, the NEP was estimated by coupling the optimized CASA model, geostatistical model of soil respiration (GSMSR) and the soil respiration–soil heterotrophic respiration ($R_s$-$R_h$) relationship model. The ground observations from ChinaFLUX were used to verify the NEP estimation accuracy. The results showed that the $R^2$ of the optimized CASA model increased from 0.411 to 0.774, and RMSE decreased from 21.425 gC·m$^{-2}$·month$^{-1}$ to 12.045 gC·m$^{-2}$·month$^{-1}$, indicating that optimizing CASA model by vegetation classification data was an effective method to improve the estimation accuracy of NEP. On this basis, the spatial and temporal distribution of NEP in China was analyzed. The research indicated that the monthly variation of NEP in China was a single peak curve with summer as the peak, which generally presented the pattern of southern region > northern region > Qinghai–Tibet region > northwest region. Furthermore, from 2001 to 2016, most regions of China showed a non-significant level upward trend, but main cropland (e.g., North China Plain and Northeast Plain) and some grassland (e.g., Ngari in Qinghai–Tibet Plateau and Xilin Gol League in Inner Mongolia) showed a non-significant-level downward trend. The study can deepen the understanding of the distribution of carbon sources/sinks in China, and provide a reference for regional carbon cycle research.

**Keywords:** NEP; CASA model; $\varepsilon_{max}$; carbon sink; spatiotemporal pattern

## 1. Introduction

Net ecosystem productivity (NEP) is an important indicator of the ecosystem carbon budget, which describes the amount of $CO_2$ in the atmosphere that an ecosystem can fix in a unit of time and represents the actual carbon capture of an ecosystem [1,2]. As China is one of the most diverse climates and ecosystems in the world, accurate estimates of NEP and analysis of spatiotemporal variation in NEP are critical in assessing the carbon balance of the Chinese terrestrial ecosystem. This information is meaningful to evaluate the carbon sequestration capacity of the ecosystems and study their carbon cycle mechanism [3–6].

NEP is the difference between net primary productivity (NPP) and soil heterotrophic respiration ($R_h$) [7]. A positive NEP means that the ecosystem stores carbon and is a carbon sink; a negative NEP means that the ecosystem releases carbon and is a carbon source [8]. Therefore, to calculate NEP, it is necessary to estimate NPP and $R_h$. Traditional NPP estimation methods include Eddy Covariance (EC) technique, the direct harvest method, biomass survey method, photosynthesis method, radiation method, chlorophyll estimation method and raw material consumption measurement method [9,10]. These methods

have high accuracy, but they are time-consuming and laborious and present difficulties in determining NPP at a wide range. Therefore, at present, macroscale (such as global scales and national scales) NPP is mainly obtained by means of remote sensing and model estimation [11]. Existing models include climate productivity models (e.g., Miami [12] and Thornthwaite [13]), physiological and ecological process models (e.g., CENTURY [14] and BIOME-BGC [15]) and light energy utilization models (e.g., C-FIX [16], CASA [17], and GLO-PEM [18]). Among them, the Carnegie Ames Stanford Approach (CASA) light energy utilization model is a simple model that can convert the absorbed photosynthetic active radiation (APAR) by vegetation and the light energy utilization rate ($\varepsilon$ (x, t)) into the calculation of the normalized difference vegetation index (NDVI), precipitation, air temperature and solar radiation. This approach can make full use of the advantages of the wide coverage and high temporal frequency obtainable from the global polar orbiting sensors such as Terra and Aqua MODIS, Spot VEGETATION, and NOAA AVHRR [19,20]. Therefore, the CASA model has become one of the mainstream models for estimating NPP, and is widely used in global and regional NPP estimation [1].

When using the CASA model to estimate NPP, two parameters should be considered. First, the maximum light use efficiency ($\varepsilon_{max}$) of global vegetation was defined as 0.389 gC·MJ$^{-1}$ in the original CASA model [21–23]. However, the value of $\varepsilon_{max}$ has always been controversial because it varies with different vegetation types [24–27]. In addition, the soil moisture submodel used to estimate the water stress coefficient ($W_\varepsilon$) involves many physical parameters and it is difficult to obtain the data. The estimation results are affected by the spatial heterogeneity of soil [28]. To solve the above problems, the monthly estimated evapotranspiration (EET) and monthly potential evapotranspiration (PET) were calculated by the regional EET model and Bouchet complementary relationship in this paper [29,30]. We use the ratio of EET and PET to calculate $W_\varepsilon$ to reduce the model parameters and simplify the estimation process [31]. Moreover, based on the study by Zhu et al. [32] and International Geosphere Biosphere Programme (IGBP) classification data, we set optimized $\varepsilon_{max}$ values for different vegetation types to improve the estimation accuracy of the CASA model.

After the calculation of NPP, it is crucial to evaluate $R_h$ to calculate NEP and evaluate its accuracy [33]. $R_h$ can be measured by direct sampling, and a wide range of data can be obtained by spatial interpolation [34]. However, due to the complexity of the soil environment and the difficulty of sampling, regions with high spatial heterogeneity will have too large of an error range for spatial interpolation with a small number of points [35]. $R_h$ can also be estimated by establishing empirical models between environmental factors (such as temperature and humidity) and measured values [36–39]. However, due to the differences in the limiting factors affecting $R_h$ in different regions (for example, water is the main factor in the northwestern arid and semiarid areas, while temperature is the main factor in the Qinghai–Tibet and northeastern regions), the method of directly using environmental factors to solve $R_h$ is not yet mature [5,40]. Moreover, $R_h$ is heterogeneous in different vegetation types, and it is difficult to accurately analyze the changes in $R_h$ through a single factor due to the comprehensive influence of the climatic environment, vegetation and soil environment [34,40,41]. Therefore, $R_h$ is usually estimated indirectly by establishing an empirical statistical model of soil respiration ($R_s$) [40–42]. Among the $R_s$ estimation models, the geostatistical model of soil respiration (GSMSR) is widely used in the large-scale quantitative analysis of $R_s$ because of its simple structure and good parameterization method [43,44]. Therefore, we can use the GSMSR to estimate the $R_s$ and $R_s$–$R_h$ relationship model to calculate $R_h$ in China.

In this paper, we will combine remote sensing data, meteorological data and soil organic carbon density (SOCD) data to estimate and analyze the NEP in China through the following steps: (1) optimize the parameter of CASA model by vegetation classification data to improve the estimation accuracy of NPP; (2) on this basis, use meteorological data and SOCD data to estimate the annual NEP and interannual NEP of Chinese terrestrial ecosystem by coupling the GSMSR and $R_s$–$R_h$ relationship model; (3) use the estimated

results to analyze the spatiotemporal distribution characteristics and change trends of NEP to provide support for carbon source and sink estimation and carbon cycle research in China.

## 2. Materials and Methods

### 2.1. Study Area and Sites Description

China has continental monsoon climate, alpine climate, arid climate and other climate types; Chinese terrain from west to east in a three-step distribution. The complex terrain and climate types form complex terrestrial ecosystems in China, as well as causing the spatial differences of carbon source and sink distribution in different regions [45]. Therefore, according to the distribution of topography and climate, we divided China into four geographical regions: the southern region (103–123°E, 22–34°N), northern region (103–135°E, 33–53°N), northwestern region (73–123°E, 37–50°N) and Qinghai–Tibet region (73–104°E, 27–40°N) (Figure 1) [46,47]. Among them, the Qinling Mountains and the Huaihe River line is the dividing line between the northern and southern regions. The Greater Khingan Mountains–Yinshan Mountains–Helan Mountains form the boundary between the northern and northwestern regions. The dividing line between Qinghai–Tibet region and northwest region, north region and south region is roughly the dividing line of the first-class ladder and the second-class ladder.

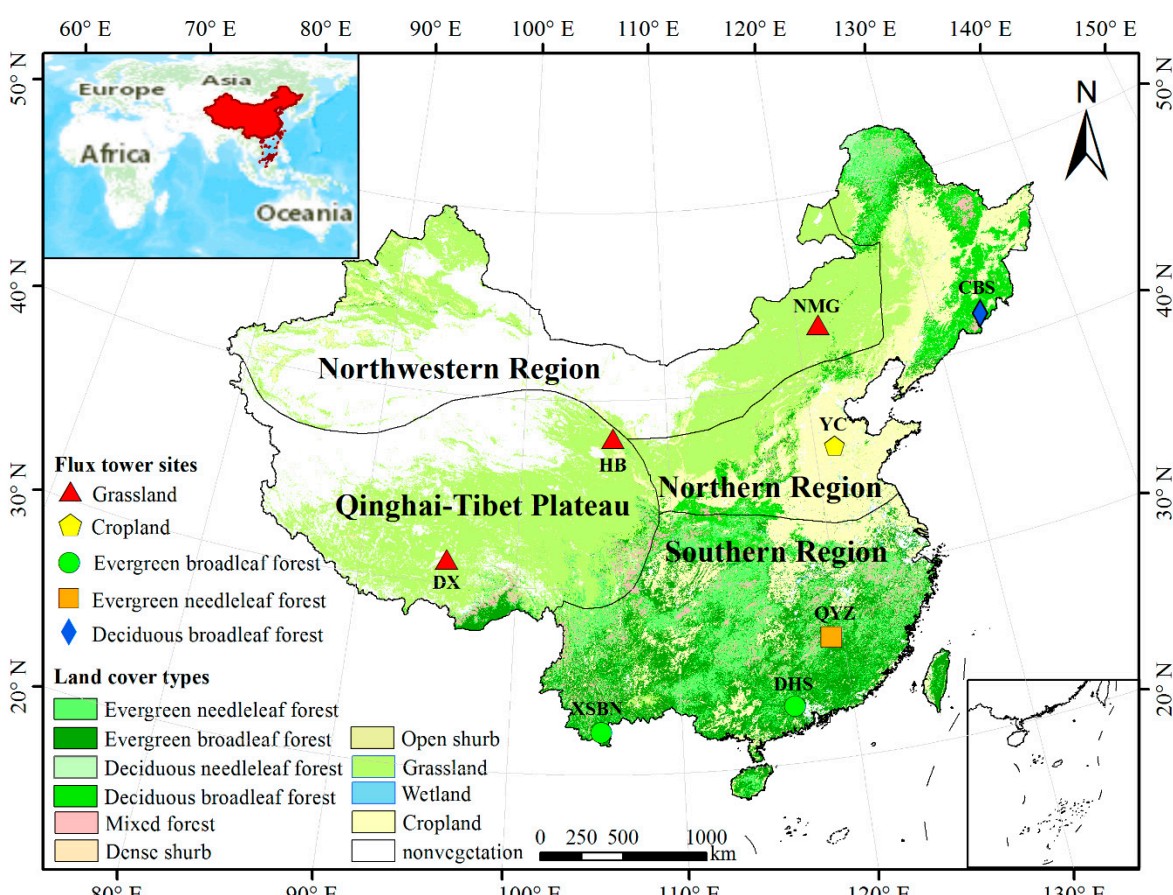

**Figure 1.** Land cover types, the locations of the eight flux tower sites and the distribution of the four geographical regions.

Eddy covariance flux towers directly measure the net ecosystem exchange (NEE) of carbon dioxide between ecosystems and the atmosphere [48]. NEE is often referred to as approximate net ecosystem productivity, which is the opposite of NEP (NEP = −NEE) [6]. We used the measurements from the Chinese Terrestrial Ecosystem Flux Research Network (ChinaFLUX) dataset for the verification of the estimation. In total, eight flux tower

sites were used in this study, namely, XSBN, DHS, QYZ, CBS, DX, HB, NMG and YC (Figure 1, Table 1). There are five vegetation types, including evergreen broadleaf forest (EBF), evergreen needle-leaf forest (ENF), deciduous broadleaf forest (DBF), grassland and cropland. The XSBN and DHS sites were characterized by the EBF vegetation type, the QYZ site was ENF, the CBS site was DBF, the DX, HB and NMG sites were grassland, and the YC site was cropland.

**Table 1.** Detailed description of sites information.

| Site Code | Site Name | Location | Climate | Data Period | Reference |
|---|---|---|---|---|---|
| XSBN | Xishuangbanna | 101°16′E 21°54′N | Temperate continental monsoon climate | January 2010–December 2010 | [48] |
| DHS | Dinghushan | 112°30′E 23°09′N | Monsoon humid climate of torrid zone | January 2010–December 2010 | [49] |
| QYZ | Qianyanzhou | 115°03′29″E 26°44′29″N | Typical subtropical monsoon climate | January 2010–December 2010 | [50] |
| CBS | Changbaishan | 128°05′45″E 42°24′9″N | Temperate continental monsoon climate | January 2010–December 2010 | [48] |
| DX | Dangxiong | 91°03′E 30°29′N | Plateau monsoon climate | January 2010–December 2010 | [51] |
| HB | Haibei | 101°19′E 37°37′N | Highland continental climate | January 2010–December 2010 | [52] |
| NMG | Neimenggu | 116°40′E 43°32′N | Temperate semi-arid continental climate | January 2010–December 2010 | [53] |
| YC | Yucheng | 116°34′E 36°50′N | Temperate semi-humid and monsoon climate | January 2010–December 2010 | [54] |

*2.2. Data Sources and Processing*

2.2.1. Land Cover Types

In this study, the MODIS land cover type product (MCD12Q1) at 500 m resolution in 2001, 2004, 2007, 2010, 2013 and 2016 was obtained from the National Aeronautics and Space Administration (NASA) (http://ladsweb.modaps.eosdis.nasa.gov/, accessed on 20 September 2021). MCD12Q1 was preprocessed by format conversion, projection conversion, image mosaic and resampling with the MODIS Reprojection Tool (MRT). We extracted the classified data from the IGBP. After that, based on the 17 land cover types classified by the IGBP classification system, we further merged them into 11 different land use types, including ENF, EBF, deciduous needle-leaf forest (DNF), DBF, mixed forest (MXF), dense shrub (DS), open shrub (OS), grassland, wetland, cropland and non-vegetation (Figure 1) [6,19,55].

2.2.2. MODIS NPP Product (MOD17A3H v006)

The MODIS NPP product (MOD17A3H v006) at a spatial resolution of 500 m was obtained from NASA (http://ladsweb.modaps.eosdis.nasa.gov/, accessed on 25 September 2021), and this product was used to compare with the accuracy of the CASA model. This product was also preprocessed by format conversion, projection conversion, image mosaic and resampling in MRT.

2.2.3. NDVI

The monthly NDVI dataset at a spatial resolution of 1 km was downloaded from the Resource and Environment Science and Data Center (RESDC) (www.resdc.cn, accessed on 15 September 2021). The dataset of Chinese monthly NDVI is based on the continuous time series of SPOT/VEGETATION NDVI satellite remote sensing data, which had undergone preprocessing at the RESDC, including atmospheric correction, radiation correction and geometric correction, and used the maximum value synthesis method to generate the monthly vegetation index datasets [56]. The dataset is widely used in the monitoring

of vegetation change, rational utilization of vegetation resources and other ecological environment related fields [57].

### 2.2.4. Meteorological Datasets

The meteorological data including the China monthly precipitation and monthly average temperature dataset, with a spatial resolution of 1 km in 2001, 2004, 2007, 2010, 2013 and 2016, were downloaded from the National Earth System Science Data Center (www.geodata.cn, accessed on 15 September 2021). The meteorological dataset was generated by the Delta spatial downscaling procedure based on the global 0.5° climate data released by the Climatic Research Unit (CRU) and the global high-resolution climate data released by WorldClim. The dataset was verified by 496 meteorological stations on the ground, which has a credible verification result [58].

### 2.2.5. DEM

A digital elevation model (DEM) is used as the input value for solar radiation area in ArcGIS 10.2 (Environmental Systems Research Institute, Inc. 2013, Redlands, CA, USA) to calculate the total solar radiation. The DEM data were downloaded from the Resource and Environment Science and Data Center (www.resdc.cn, accessed on 20 September 2021). The spatial distribution data are derived from the Shuttle Radar Topography Mission (SRTM) of the United States, which is based on the resampling of SRTM v4.1 data, and its spatial resolution is 1 km.

### 2.2.6. SOCD

Soil organic carbon density (SOCD) data were used as the input value of the GSMSR to calculate the $R_s$ of the study area. According to the classification of secondary vegetation, we chose the SOCD of different vegetation types at a depth down to 20 cm [59]. Among them, the value for coniferous forest is 3.770 kg·m$^{-2}$, broadleaf forest is 4.700 kg·m$^{-2}$, the value for MXF is the average of coniferous forest and broadleaf forest with 4.235 kg·m$^{-2}$, shrub is 2.560 kg·m$^{-2}$, grassland is 1.820 kg·m$^{-2}$, cropland is 2.560 kg·m$^{-2}$, other vegetation types are 0 [59].

### *2.3. Research Methods*

#### 2.3.1. Estimation of NPP Based on CASA Model

In 1993, Potter et al. [23] proposed the CASA model, which takes remote sensing data as input data, combines environmental variables (temperature, moisture, soil) and vegetation physiological parameters, and uses the product of APAR and light energy utilization ($\varepsilon(x, t)$) to represent NPP. The expression of NPP is as follows [23]:

$$\mathrm{NPP}(x, t) = \mathrm{APAR}(x, t) \times \varepsilon(x, t), \tag{1}$$

where $\mathrm{APAR}(x, t)$ is the photosynthetically active radiation absorbed by pixel x in month t (MJ·m$^{-2}$·month$^{-1}$) and $\varepsilon(x, t)$ is the actual light energy utilization of pixel x in month t (gC·MJ$^{-1}$).

- APAR

The spectral range of solar radiation is 115–5000 nm, but the solar radiation in this range cannot be completely absorbed by plants. Vegetation can actually use solar radiation in the wavelength range of 380–710 nm for photosynthesis, which is called photosynthetic active radiation (PAR). APAR absorbed by vegetation can be expressed by the total solar radiation and the fraction of photosynthetic active radiation (FPAR). These equations of APAR and FPAR are all taken from the Potter et al. paper. The expression of APAR is as follows [23]:

$$\mathrm{APAR}(x, t) = 0.5 \times \mathrm{SOL}(x, t) \times \mathrm{FPAR}(x, t), \tag{2}$$

where SOL(x, t) is the total solar radiation of pixel x in month t (MJ·m$^{-2}$·month$^{-1}$); FPAR(x, t) refers to the absorption ratio of vegetation to incident PAR; and 0.5 refers to the ratio of solar effective radiation to total solar radiation. In Equation (2), FPAR is expressed as:

$$FPAR(x, t) = \alpha \, FPAR_{SR} + (1 - \alpha) \, FPAR_{NDVI}, \tag{3}$$

where $\alpha = 0.5$. The calculation formula for $FPAR_{SR}$ and $FPAR_{NDVI}$ are as follows:

$$FPAR_{SR} = \frac{SR(x, t) - SR_{(i, min)}}{SR_{(i, max)} - SR_{(i, min)}} \times (FPAR_{max} - FPAR_{min}) + FPAR_{min}, \tag{4}$$

$$SR(x, t) = \frac{1 + NDVI(x, t)}{1 - NDVI(x, t)}, \tag{5}$$

$$FPAR_{NDVI} = \frac{NDVI(x, t) - NDVI_{(i,min)}}{NDVI_{(i, max)} - NDVI_{(i, min)}} \times (FPAR_{max} - FPAR_{min}) + FPAR_{min}, \tag{6}$$

where $SR_{(i, min)}$, $SR_{(i, max)}$, $NDVI_{(i, min)}$, $NDVI_{(i, max)}$, $FPAR_{min}$ and $FPAR_{max}$ represent the minimum and maximum values of the "simple ratio" (SR), NDVI and FPAR for the i-th vegetation type, respectively. Among them, $FPAR_{min}$ and $FPAR_{max}$ are independent of vegetation type, with 0.001 and 0.950, respectively.

- Light energy utilization efficiency ($\varepsilon(x, t)$) and its optimization

$\varepsilon(x, t)$ refers to the efficiency of green vegetation absorbing light energy and converting it into organic carbon in a certain period of time [59]. This value is another important parameter for estimating NPP, which is affected by temperature, moisture and $\varepsilon_{max}$ in the real environment. The expression of $\varepsilon(x, t)$ is as follows:

$$\varepsilon(x, t) = T_{\varepsilon 1}(x, t) \times T_{\varepsilon 2}(x, t) \times W_{\varepsilon}(x, t) \times \varepsilon_{max}, \tag{7}$$

where $T_{\varepsilon 1}(x, t)$ and $T_{\varepsilon 2}(x, t)$ indicate the stress effect of low temperature and high temperature on the light energy utilization efficiency, respectively. $W_{\varepsilon}(x, t)$ is the influence coefficient of water stress. $\varepsilon_{max}$ is the maximum utilization of light energy under ideal conditions (gC·MJ$^{-1}$). The expression of $T_{\varepsilon 1}(x, t)$ and $T_{\varepsilon 2}(x, t)$ are as follows:

$$T_{\varepsilon 1}(x, t) = 0.8 + 0.02 \times T_{opt}(x) - 0.0005 \times [T_{opt}(x)]^2, \tag{8}$$

$$T_{\varepsilon 2}(x, t) = 1.1814 / \{1 + \exp[0.2 \times (T_{opt}(x) - 10 - T(x, t))]\} \times 1 / \{1 + \exp[0.3 \times (-T_{opt}(x) - 10 + T(x, t))]\} \tag{9}$$

where $T_{opt}(x)$ is the optimum temperature for plant growth, which is defined as the average temperature (°C) of the month when the NDVI value reaches the highest in a certain area in a year; when the average temperature $T(x, t)$ in a month is less than or equal to $-10$ °C, the value of $T_{\varepsilon 1}(x, t)$ is taken as 0; when the average temperature $T(x, t)$ in a month is 10 °C higher or 13 °C lower than the optimum temperature, the value of $T_{\varepsilon 2}(x, t)$ in a month is equal to half of the $T_{\varepsilon 2}(x, t)$ value when the monthly average temperature $T(x, t)$ is the optimum temperature $T_{opt}(x)$.

$W_{\varepsilon}$ reflects the effect of available water conditions on $\varepsilon$, which ranges from 0.5 (under extreme drought conditions) to 1 (under very humid conditions). The calculation of $W_{\varepsilon}$ is as follows:

$$W_{\varepsilon}(x, t) = 0.5 + 0.5 \times EET \, PET, \tag{10}$$

where EET is regional estimated evapotranspiration (mm) and PET is regional potential evapotranspiration (mm).

Since it is difficult to obtain EET and PET in reality, the EET is calculated by the regional estimated evapotranspiration model of Zhou and Zhang [31]. The PET is calculated using

climate indicators used in the Holdridge life zone diagram [60]. The calculation of EET is as follows:

$$EET = \frac{P \cdot R_n \left(P^2 + R_n{}^2 + P \cdot R_n\right)}{(P + R_n) \left(P^2 + R_n{}^2\right)}, \tag{11}$$

$$R_n = (PET \cdot P)^{0.5} \cdot \left[0.369 + 0.598 \, (PET \, / \, P)^{0.5}\right], \tag{12}$$

where P is the precipitation of pixel x in month t and $R_n$ is the net solar radiation of pixel x in month t. The calculation of PET is as follows [60]:

$$PET = BT \times 58.93, \tag{13}$$

$$BT = \frac{1}{12} \sum_1^{12} T_i, \tag{14}$$

where BT is the biological temperature and $T_i$ is the monthly average temperature of more than 0 °C. The temperature above 30 °C is calculated as 30 °C, and that below 0 °C is calculated as 0 °C.

The $\varepsilon_{max}$ refers to the light energy conversion rate of vegetation in the ideal state. In original CASA model, the $\varepsilon_{max}$ of global vegetation was set as 0.389 gC·MJ$^{-1}$ [26,61,62]; however, studies have shown that the $\varepsilon_{max}$ of different vegetation types is different. Therefore, it is important to reasonably determine the model parameter $\varepsilon_{max}$ to estimate NPP accurately with CASA model. In this paper, referring to the study of Zhu et al. and Yu et al. [32,63,64], we revalue the parameter $\varepsilon_{max}$ as Table 2.

**Table 2.** Maximum light energy utilization ($\varepsilon_{max}$) of typical vegetation types in China (gC·MJ$^{-1}$).

| Vegetation Types | $\varepsilon_{max}$ |
| --- | --- |
| ENF | 0.476 |
| EBF | 0.985 |
| DNF | 0.485 |
| DBF | 0.692 |
| MXF | 0.768 |
| DS | 0.429 |
| OS | 0.429 |
| Grassland | 0.542 |
| Cropland | 0.542 |
| Other | 0.389 |

2.3.2. Estimation of Soil Respiration

In this paper, combined with meteorological data and SOCD data, we use the GSMSR to estimate $R_s$, and then calculate $R_h$ in China according to the $R_s$–$R_h$ relationship model.

- Geostatistical model of soil respiration (GSMSR)

The total soil respiration will be estimated using the GSMSR. The GSMSR is a global statistical model for estimating soil respiration, which uses the monthly average temperature, monthly precipitation and SOCD as input data [43]. First, we input temperature data to determine $R_s$ [43]:

$$R_s = R_0 \times e^{bt}, \tag{15}$$

where $R_s$ is the instantaneous soil respiration rate $\left(umol·m^{-2}·s^{-1}\right)$, $R_0$ is the soil respiration rate $\left(umol·m^{-2}·s^{-1}\right)$ at the reference temperature of 0 °C, and t is the actual air temperature.

The relationship between global soil respiration and temperature and precipitation can be expressed as follows [43]:

$$R_{s, \, monthly} = R_0 \times e^{QT} \times P \, / \, (P + K), \tag{16}$$

where $R_0 = 1.250$, $Q = 0.055$, $K = 4.250$.

Since the respiration characteristics of different ecosystems in the same site cannot be fully obtained, there are still many systematic errors in the model. Considering the strong correlation between residuals and SOCD, SOCD is taken as a parameter of the calculation model. Therefore, the model can be expressed as follows [43]:

$$R_{s,\,monthly} = (R_{D_s=0} + M \times D_s) \times e^{QT} \times (P + P_0) \, / \, (P + K), \qquad (17)$$

where $D_s$ is the SOCD of soil at a depth of 20 cm; $R_{D_s=0}$ is the monthly average soil respiration when SOCD is 0, and M is the parameter. Since the assumption of "zero rainfall and zero exhalation" in the original model is not in line with the actual ecological situation, the parameter $P_0$ is added to the formula to consider the water capacity retained in the soil.

The relationship between the spatial variation characteristics of the $Q_{10}$ value and air humidity in China can be expressed as follows:

$$Q_{10} = \ln \alpha e^{\beta T}, \qquad (18)$$

In the formula, $\alpha$ and $\beta$ are fitted parameters, and the corrected monthly soil respiration can be expressed as follows:

$$R_{s,\,monthly} = (R_{D_s=0} + M \times D_s) \times e^{\ln \alpha e^{\beta T} T/10} \times (P + P_0) \, / \, (P + K), \qquad (19)$$

where $R_{D_s=0} = 0.588 \; (gC \cdot m^{-2} \cdot month^{-1})$, $M = 0.118$, $\alpha = 1.830$, $\beta = 0.006$, $P_0 = 2.972$, $K = 5.657$.

- $R_s$–$R_h$ relationship mode

According to the $R_s$ calculated by the above GSMSR, $R_h$ is calculated according to the empirical relationship between $R_s$ and $R_h$. In this paper, referring to the research of Shi [65], we use the $R_s$–$R_h$ relationship model to calculate the $R_h$ of China. The relationship model is based on the real $R_s$ and $R_h$ data of different time periods and different locations in China, and its calculation formula is as follows [65]:

$$R_h = -0.0009 R_s{}^2 + 0.6011 R_s + 4.8874, \qquad (20)$$

where $R_s$ is monthly soil respiration $\left( gC \cdot m^{-2} \cdot month^{-1} \right)$

### 2.3.3. Verification Method of NEP Estimation Results

The measured net ecosystem exchange (NEE) for verification was downloaded from National Ecological Science Data Resource Center (www.cnern.org.cn, accessed on 28 September 2021), and the data were subjected to the quality control and data processing of the ChinaFLUX technical system standardization [49]. The technical system is based on the technical processes which was widely adopted and accepted in the field of global flux observation and research. Among them, the standardization of data quality control including raw data analysis [66], ultrasonic virtual temperature correction [67], coordinate rotation [68], WPL correction [69], frequency correction loss [70], canopy stored item correction [71], the steady-state test and closure turbulence integral characteristics [66], energy evaluation [72], etc. Data processing includes interpolation of missing data by means of mean diurnal variation method, nonlinear regression method and marginal distribution sampling method, and separation of $CO_2$ flux data by means of marginal distribution sampling method, etc. After processing, the monthly measured NEE data of the site is finally stored in Excel format for users.

The verification of this study was carried out based on the above measured NEE. First, we converted the directly measured NEE into the measured NEP value by taking a negative number. Then, according to the global positioning system (GPS) coordinates of each station, we extracted the estimated NEP value of $1 \times 1$ km pixels of each station. Finally, coefficient

of determination ($R^2$) and the root mean square errors (RMSEs) were used as precision indexes to calculate the linear relationship between the estimated value of NEP and the measured value. In addition, due to the relatively complete ground observation data in 2010, this paper uses the data of that year for verification.

2.3.4. Anomaly NEP (ANEP)

The anomaly NEP (ANEP) was used to analyze historical changes of the NEP and was determined using the following equation [47,73]:

$$\text{ANEP} = (\text{NEP}_i - \text{NEP}_{ave}) / \text{NEP}_{ave}, \tag{21}$$

where ANEP is the anomaly NEP, $\text{NEP}_i$ is the NEP value during a specific period, and $\text{NEP}_{ave}$ is the average NEP value during the studied period. A positive ANEP indicates that the NEP value of the current year is higher than the average annual value, while a negative ANEP indicates that the NEP value of the current year lower than that the average annual value.

2.3.5. NEP Variation Trend Analysis

To understand the overall variation trends of NEP from 2001 to 2016, the following equation was used to calculate the trend rate [47,73]:

$$Slope = \frac{n \times \sum_{i=1}^{n} x_i t_i - \sum_{i=1}^{n} x_i \sum_{i=1}^{n} t_i}{n \times \sum_{i=1}^{n} t_i^2 - \left(\sum_{i=1}^{n} t_i\right)^2}, \tag{22}$$

where $t_i$ is the serial number from 2001 to 2016 (with a 3-year interval) (1–6), $n$ is the total length of the time series ($n = 7$), and $x_i$ is the NEP value in year $i$. In addition, *slope* > 0 indicated that the NEP value showed an upward trend during the research period; otherwise, it showed a downward trend.

**3. Results**

*3.1. The Analysis of NPP Based on the CASA Model*

3.1.1. NPP Estimation Results

Based on the solar radiation data, DEM data, NDVI data, temperature, humidity and other meteorological data, the NPP values were calculated by using the original and optimized CASA model respectively. The NPP of China's terrestrial ecosystems in 2010 estimated by optimized CASA model is shown in Figure 2.

Figure 2 highlights the seasonal characteristics of NPP in China—generally, it is high in summer and low in winter, but there are differences in various geographical regions. NPP shows the distribution pattern of southern region > northern region > Qinghai–Tibet region > northwestern region. The NPP values in the Qinghai–Tibet region, northwestern region and northern region are low in spring and winter, while NPP in the southern region remains at high values in late spring and early winter. Due to good hydrothermal conditions, the vegetation in the southern region grows vigorously and has the highest NPP of 623.907 gC·m$^{-2}$·a$^{-1}$. However, the Qinghai–Tibet region and northwestern region are affected by alpine and arid climates, which are not conducive to vegetation growth, resulting in scarce vegetation and low NPP. The NPPs of the Qinghai–Tibet region and northwestern region are 183.165 gC·m$^{-2}$·a$^{-1}$ and 102.540 gC·m$^{-2}$·a$^{-1}$, respectively.

Further analysis shows that the NPP values of various types of vegetation are different. The values of each typical vegetation type are arranged in the following order: EBF > ENF > DBF > DNF. Among them, the NPP of EBF was the highest, with a value of 811.981 gC·m$^{-2}$·a$^{-1}$ The NPP of grassland was the lowest, with a value of 183.444 gC·m$^{-2}$·a$^{-1}$ This result is consistent with the research of Zhu et al. [74] and conforms to the range of measured NPP, which shows that the results are reliable in general.

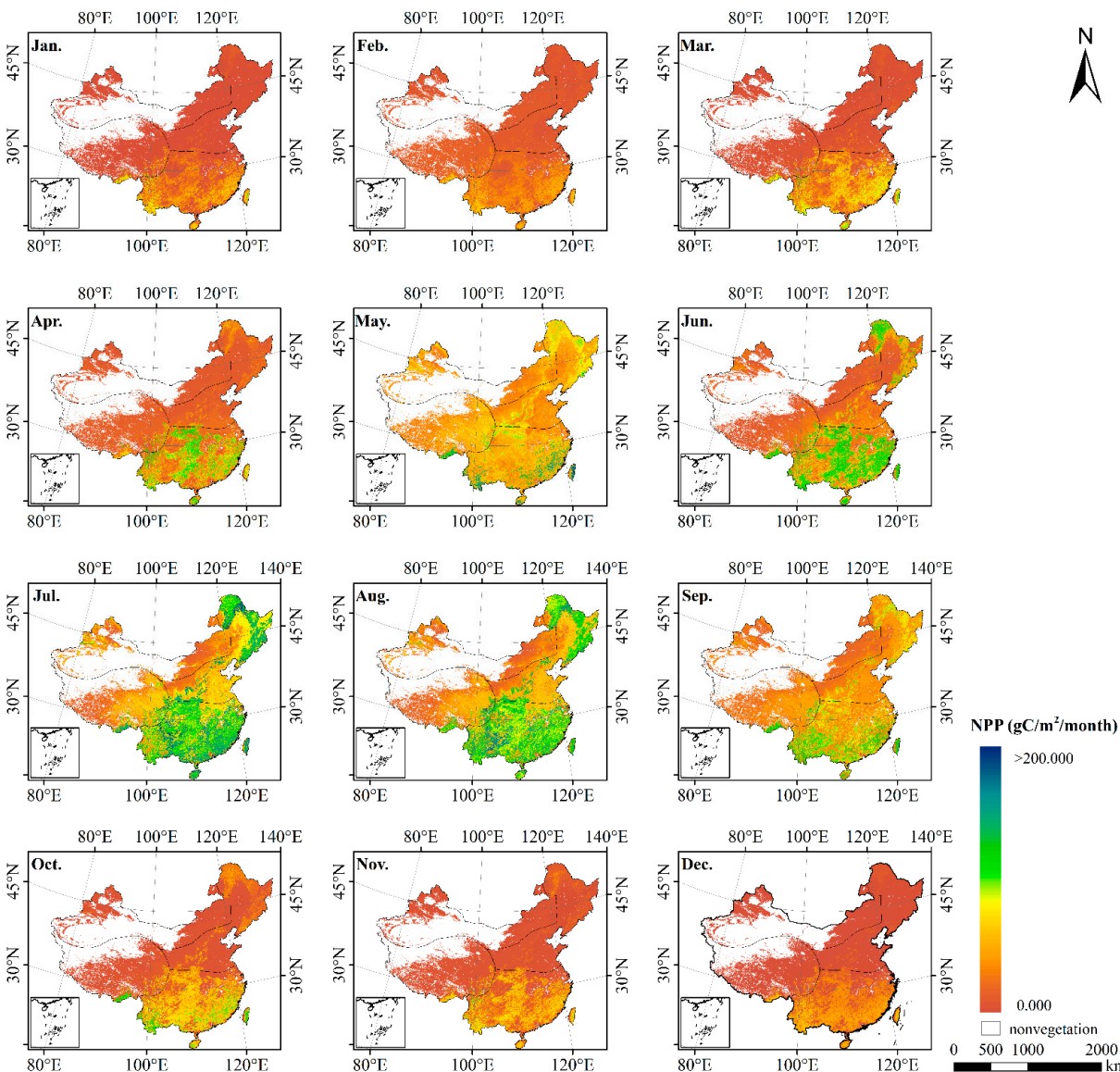

**Figure 2.** Spatial pattern of monthly NPP estimated over China during 2010 by the improved CASA model.

### 3.1.2. The Reliability Analysis of Estimated NPP

Since there are no ground acquisition data of NPP, we cannot directly verify the estimation results of the model. Nevertheless, MOD17A3H data can be used for indirect verification. MOD17A3H is an NPP data product provided by NASA's EOS/MODIS. This product has a wide range of applications in NPP research at different spatial scales, but it is available only as an annual product [75]. In this study, the monthly NPP estimation results obtained by the improved CASA model were used to calculate the annual average NPP and perform a comparative analysis (Figure 3, Table 3). Figure 3 shows that the same trend occurred between the estimated NPP results of this paper and MOD17A3H in northern region, northwestern region and Qinghai–Tibet region, but slightly different in southern region. Specifically, the NPP of Yunnan, Guangxi Zhuang Autonomous Region and Guangdong Province is slightly lower than that of MOD17A3H, while the NPP of Fujian, Zhejiang and Jiangxi Province is slightly higher than that of MOD17A3H.

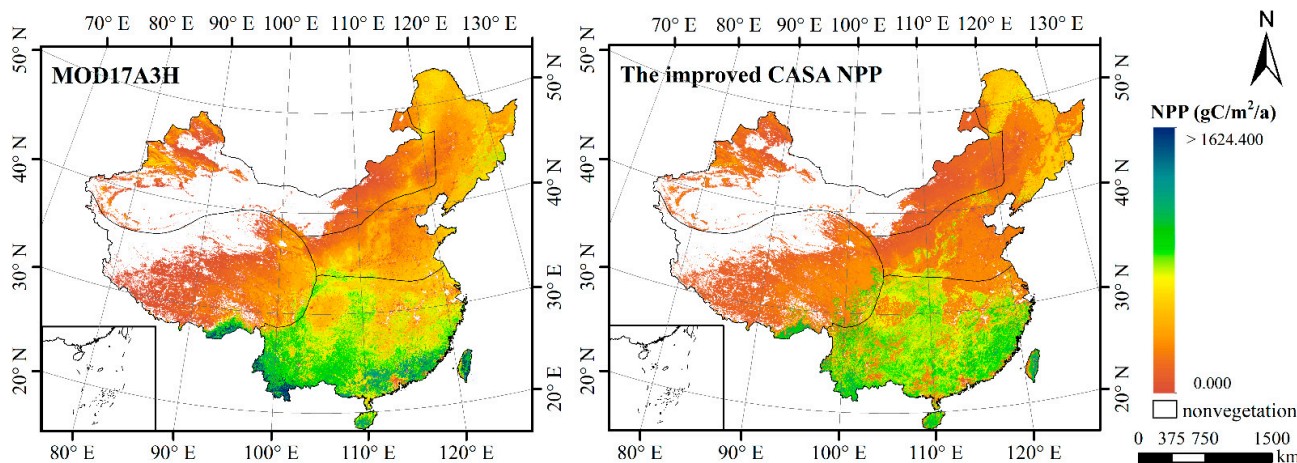

**Figure 3.** Comparison between MOD17A3H and the improved CASA NPP results.

**Table 3.** Comparison of NPP mean values of different vegetation types based on MOD17A3H and optimized CASA model.

| Vegetation Types | MOD17A3H (gC·m$^{-2}$·a$^{-1}$) | Optimized CASA Model (gC·m$^{-2}$·a$^{-1}$) | Percentage Deviation |
|---|---|---|---|
| ENF | 705.367 | 701.637 | 0.529% |
| EBF | 767.051 | 811.981 | 5.857% |
| DNF | 477.025 | 340.510 | 28.618% |
| DBF | 565.624 | 516.450 | 8.694% |
| MXF | 700.982 | 525.496 | 25.034% |
| DS | 332.307 | 366.007 | 10.141% |
| OS | 53.462 | 110.077 | 105.898% |
| grassland | 191.985 | 183.444 | 4.449% |
| wetland | 168.897 | 119.056 | 29.510% |
| cropland | 413.199 | 270.829 | 34.456% |

Table 3 is the NPP of different vegetation types based on MOD17A3H and the optimized CASA model, which shown that the percentage deviation is smaller for ENF, grassland, EBF and DBF, with deviations of 0.529 percent, 4.449 percent, 5.857 percent and 8.694 percent respectively, while the percentage deviation is larger for OS and cropland, with deviations of 105.898 percent and 34.456 percent, respectively. Among them, there are two reasons for this difference. On the one hand, the MOD17A3H product is based on the Biome-BGC model, which lacks the module for estimating crop carbon flux, leading to a relatively large error in the NPP of cropland. On the other hand, due to the complex growth process of crops, they are greatly affected by manmade factors (irrigation, fertilization, etc.), which results in great uncertainty in NPP.

### 3.2. The NEP Estimation Results Based on Coupling Model

3.2.1. NEP Estimation Results

The NPP value can be estimated by CASA model. However, NPP value cannot directly characterize carbon sinks. It is necessary to subtract soil heterotrophic respiration to obtain NEP value that can characterize carbon sinks [76]. Therefore, the study coupled the CASA model, GSMSR and $R_s$–$R_h$ relationship model to estimate the NEP value. Combined with meteorological data and SOCD data, we used the GSMSR to estimate $R_s$, and then calculate $R_h$ in China according to the $R_s$–$R_h$ relationship model. The spatial and temporal distribution of $R_h$ is shown in Figure 4.

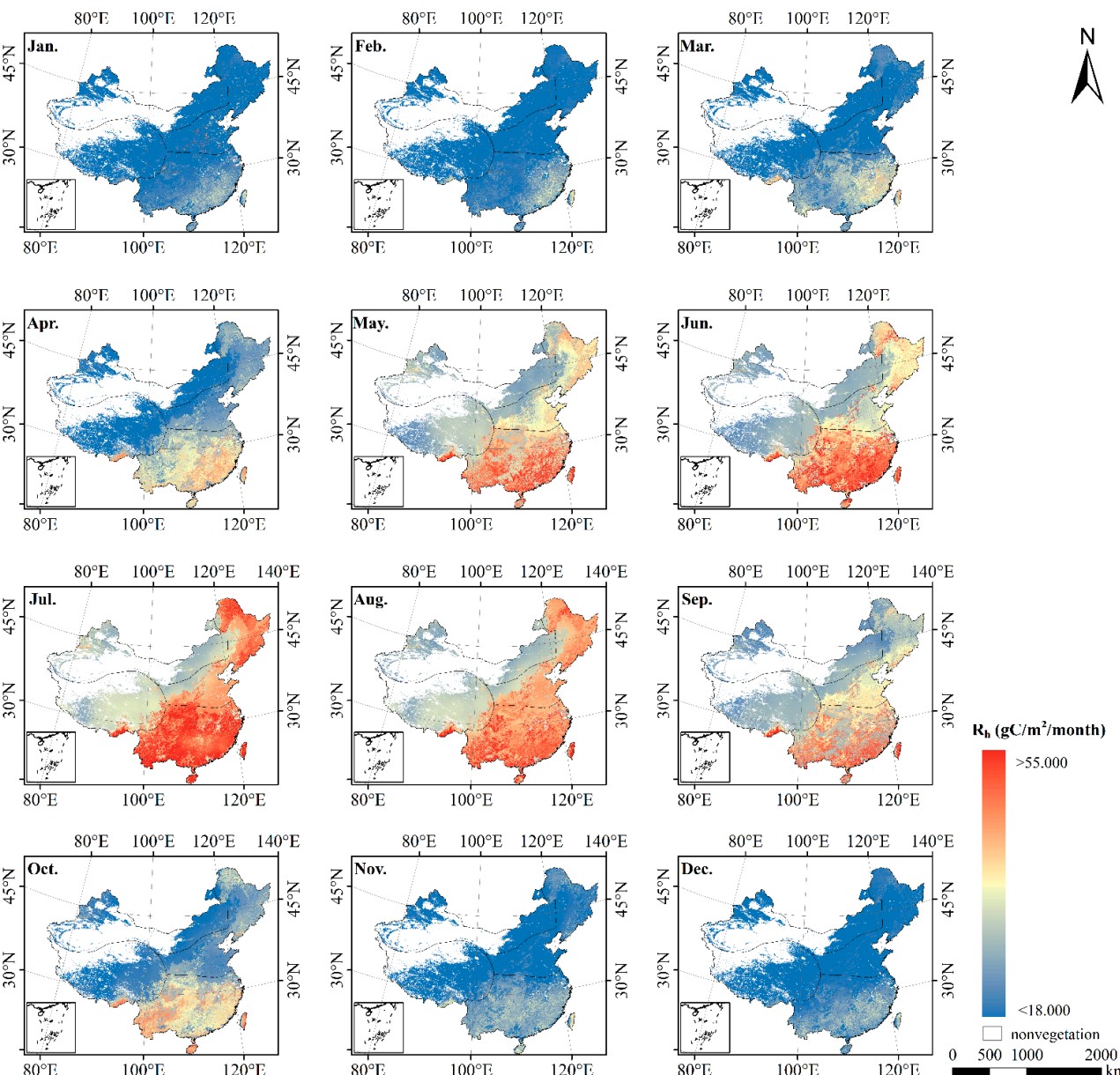

**Figure 4.** Spatial pattern of the monthly $R_h$ over China during 2010 by the GSMSR and $R_s$–$R_h$ relationship model.

As shown in Figure 4, $R_h$ in space presented the characteristics of high in the south and low in the north, with high values in the east and low values in the west. The specific pattern was southern region > northern region > Qinghai–Tibet region > northwestern region, and the $R_h$ in each region was 409.887 gC·m$^{-2}$·a$^{-1}$, 303.972 gC·m$^{-2}$·a$^{-1}$, 252.029 gC·m$^{-2}$·a$^{-1}$ and 232.537 gC·m$^{-2}$·a$^{-1}$, respectively. Seasonally, the average $R_h$ in summer was 38.220 gC·m$^{-2}$·month$^{-1}$, which was higher than that in winter.

Combined with the NPP and $R_h$ above, NEP was obtained by subtracting $R_h$ from NPP. Figure 5 presents the spatial distribution of NEP in terrestrial ecosystems in China in 2010. The results revealed that the spatial distribution of NEP was the same as that of NPP, represented by southern region > northern region > Qinghai–Tibet region > northwestern region. The NEP values in the southern region and northern region were both positive, 216.680 gC·m$^{-2}$·a$^{-1}$ and 19.195 gC·m$^{-2}$·a$^{-1}$, respectively, indicating that the southern and northern regions were overall carbon sinks. However, the NEPs of the Qinghai–Tibet region and northwestern regions were negative at −39.045 gC·m$^{-2}$·a$^{-1}$ and −95.872 gC·m$^{-2}$·a$^{-1}$, respectively, indicating that these areas were basically carbon sources.

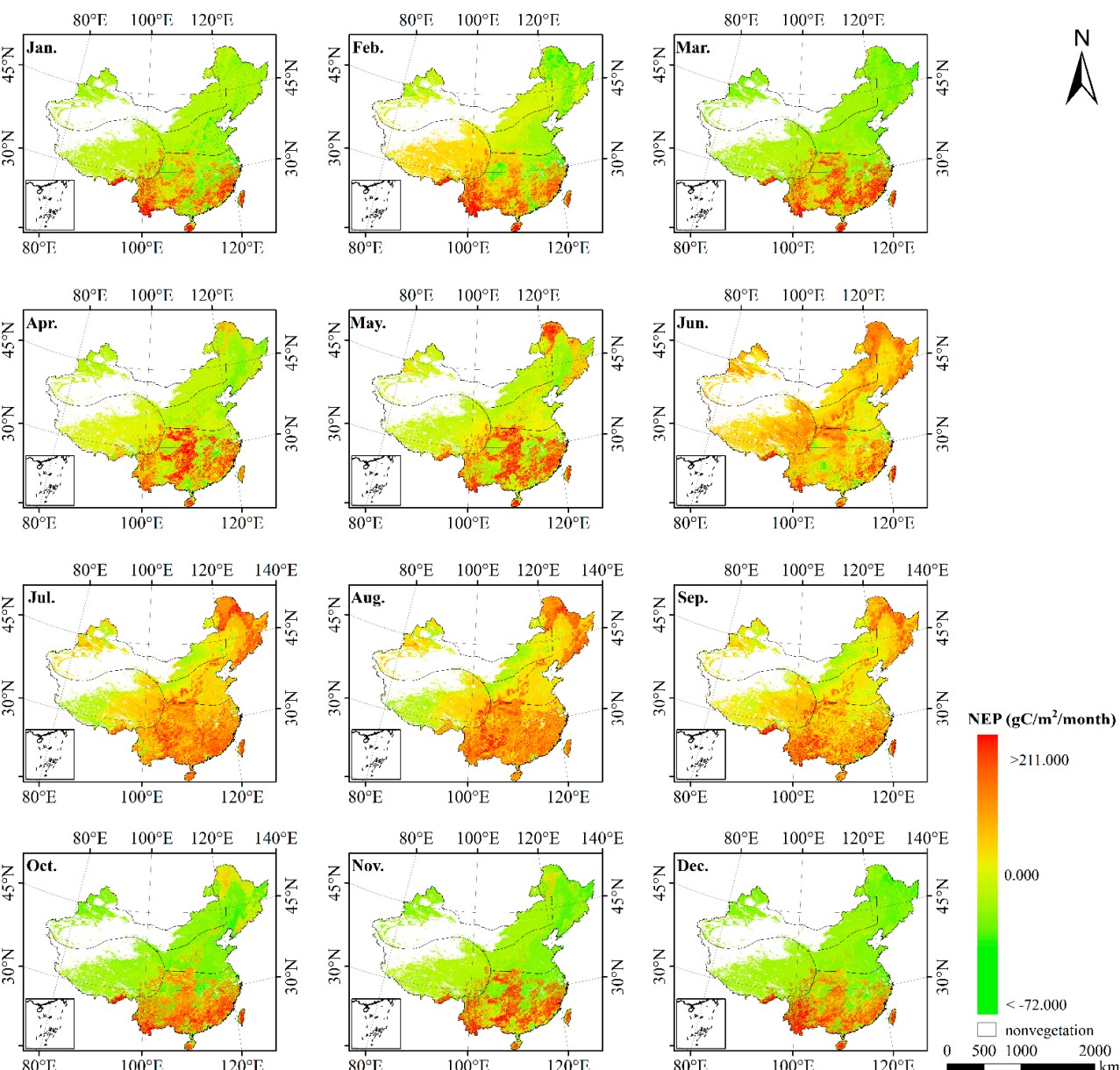

**Figure 5.** Spatial pattern of monthly NEP estimated over China during 2010 (from January to June) by the CASA model and GSMSR model.

### 3.2.2. The Accuracy of Estimated NEP

According to GPS longitude and latitude, we matched the ground observed NEP values and estimated values which included the original CASA model with the $\varepsilon_{max}$ of 0.389 gC·MJ$^{-1}$ (Figure 6a) and the optimized CASA model with $\varepsilon_{max}$ of the different vegetation types (Figure 6b). We then evaluated the accuracy with the R$^2$ and RMSEs.

In general, the estimation accuracy of the CASA model, which adopt the vegetation classification of the $\varepsilon_{max}$ parameter, was higher than that of the original CASA model. The R$^2$ increased from 0.411 to 0.774, and the RMSE decreased from 21.425 gC·m$^{-2}$·month$^{-1}$ to 12.045 gC·m$^{-2}$·month$^{-1}$ In addition, the data of the sample point at the lower left corner in Figure 6a exhibited a large deviation in the original CASA model, but it performed normally in the optimized model. The impact of removing this point on the results was further evaluated. After removing this point, the R$^2$ and RMSE of original CASA model were 0.415 and 21.363 gC·m$^{-2}$·month$^{-1}$, and that of the optimized model were 0.773 and 12.034 gC·m$^{-2}$·month$^{-1}$, respectively. Thus, the overall result of the accuracy verification

remained basically unchanged. The results indicated that the classification of vegetation could effectively improve the accuracy of the CASA model.

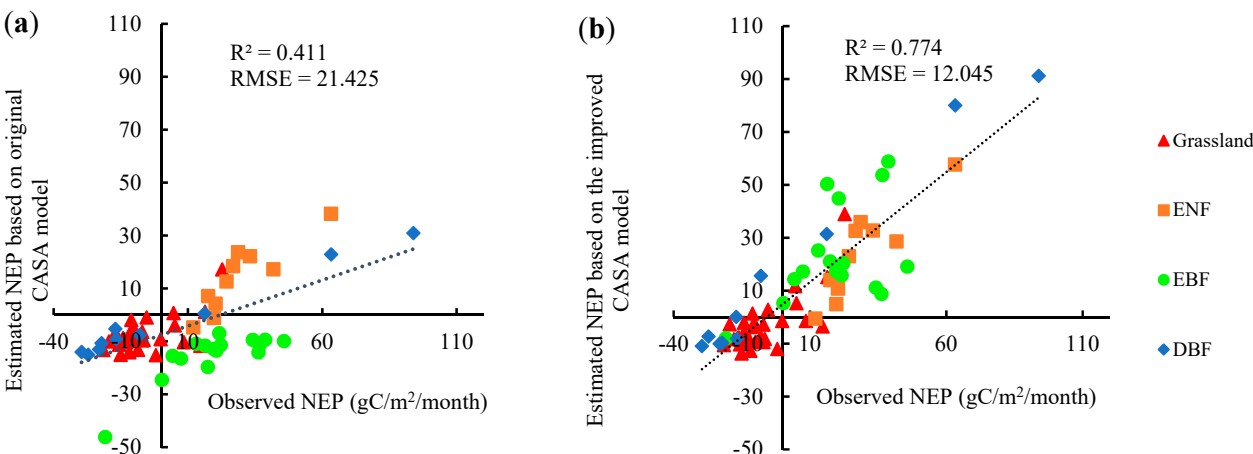

**Figure 6.** The observed NEP VS the estimated NEP based on the CASA model. (**a**) $\varepsilon_{max}$ using vegetation classification; (**b**) $\varepsilon_{max}$ of fixed value 0.389 (gC·MJ$^{-1}$).

Moreover, as shown in Figure 6a, NEP was generally underestimated by the original CASA model. The reason for this result was that according to the actual research results [32], the $\varepsilon_{max}$ of each vegetation type was higher than 0.389 gC·MJ$^{-1}$ in most cases (details in Table 2). Therefore, to accurately estimate the NEP, it was very important to estimate NPP according to the different vegetation types.

Further analysis showed that using the $\varepsilon_{max}$ in Table 2 could better improve the estimation accuracy of the CASA model, but it showed different effects on different ecosystems. The CASA model exhibited obvious improvements for the ENF, EBF and DBF ecosystems, as the RMSEs for these ecosystems decreased from the values in the original CASA model from 16.363 gC·m$^{-2}$·month$^{-1}$ to 8.282 gC·m$^{-2}$·month$^{-1}$, 36.558 gC·m$^{-2}$·month$^{-1}$ to 17.698 gC·m$^{-2}$·month$^{-1}$, and 25.810 gC·m$^{-2}$·month$^{-1}$to 15.835 gC·m$^{-2}$·month$^{-1}$, respectively. However, the CASA model exhibited a relatively weak improvement effect on grassland ecosystems, showing that the RMSE of grassland decreased from the original value of 8.519 gC·m$^{-2}$·month$^{-1}$ to 8.185 gC·m$^{-2}$·month$^{-1}$, which indicated that grassland ecosystems were less sensitive than others to $\varepsilon_{max}$.

*3.3. Analysis of Spatiotemporal Variation of NEP in China*

3.3.1. The Monthly Variation of NEP in China

The NEP monthly variation in each region and each vegetation was analyzed (Figure 7). The monthly variation of NEP in China showed a single peak curve (Figure 7a). NEP increased sharply from May to July, reaching a peak of 17.703 gC·m$^{-2}$·month$^{-1}$ in July and then decreased gradually, reaching the lowest values of −6.290 gC·m$^{-2}$·month$^{-1}$ in October. The average NEP in summer was 13.943 gC·m$^{-2}$·month$^{-1}$ in summer and −3.526 gC·m$^{-2}$·month$^{-1}$ in winter, indicating that it was a carbon source in winter and a carbon sink in summer (Figure 7a).

There were regional differences in the monthly variation of NEP. The NEP was positive in each month in the southern region (i.e., it was a carbon sink throughout the year), while that in the northern region was positive only from June to September (Figure 7b). NEP in the northern region, northwestern region and Qinghai–Tibet region maintained the same trend in early spring, late autumn and the whole winter, which were all carbon sources. In addition, the NEP in the southern region and northern region reached a peak in July, while that in the northwestern region and Qinghai–Tibet region reached a peak in June.

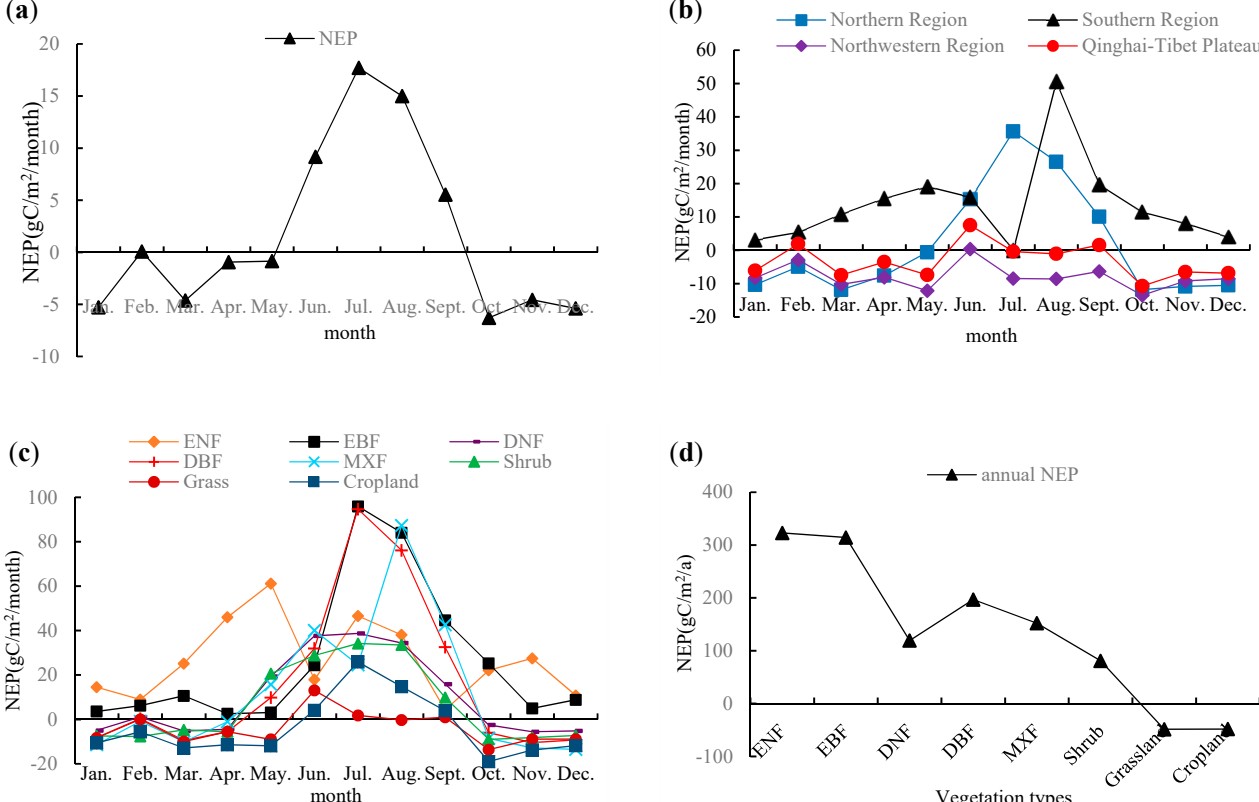

**Figure 7.** Annual analysis results of NEP in China. (**a**) The monthly average NEP of the whole country; (**b**) The monthly average NEP of different regions; (**c**) The monthly average NEP of each vegetation type; (**d**) Annual average NEP of each vegetation type.

The monthly variation of NEP of various vegetation was also different. The NEP values of ENF and EBF were positive in each month, which showed a carbon sink in general. However, the DNF, DBF, MXF, shrub, grassland and cropland ecosystems featured carbon sinks in summer and carbon sources in spring and winter (Figure 7c). Further analysis showed that from the annual average, the NEP of forest was greater than that of shrub, followed by grassland and cropland. As for forest, the NEP of evergreen forest was greater than deciduous forest and mixed forest (Figure 7d). The NEP of ENF, EBF, DNF, DBF, MXF and shrub were greater than 0, indicating that their annual values corresponded to carbon sink on the whole; the NEP of grassland and cropland were $-48.568$ gC·m$^{-2}$·a$^{-1}$ and $-48.147$ gC·m$^{-2}$·a$^{-1}$, respectively, indicating that their annual values corresponded to carbon sources on the whole (although the NPP values of grassland and cropland were positive, the NEP was less than 0 after considering the effect of $R_h$).

### 3.3.2. The Interannual Variation of NEP in China

- The spatiotemporal variation characteristics of NEP from 2001 to 2016

Due to the vigorous growth of vegetation in summer, the accumulated productivity in summer is relatively large, which can better represent the NEP level of the year. Therefore, this paper analyzes the average summer NEP from 2001 to 2016 (with a three-year interval). The spatial distribution was shown in Figures 8 and 9.

As shown in the figures, the NEP had obvious spatial heterogeneity. The summer NEP in most parts of China was greater than 0, indicating that the vast majority of places were carbon sinks. Among them, the high NEP values were in the forest areas of the Greater Khingan Mountains and Taihang Mountains in Northeast China and in the south of China. The vegetation in Northeast China is mostly coniferous forest, while the vegetation in South

China is mostly evergreen broad-leaved forest. Therefore, the vegetation has more carbon sinks and higher NEP.

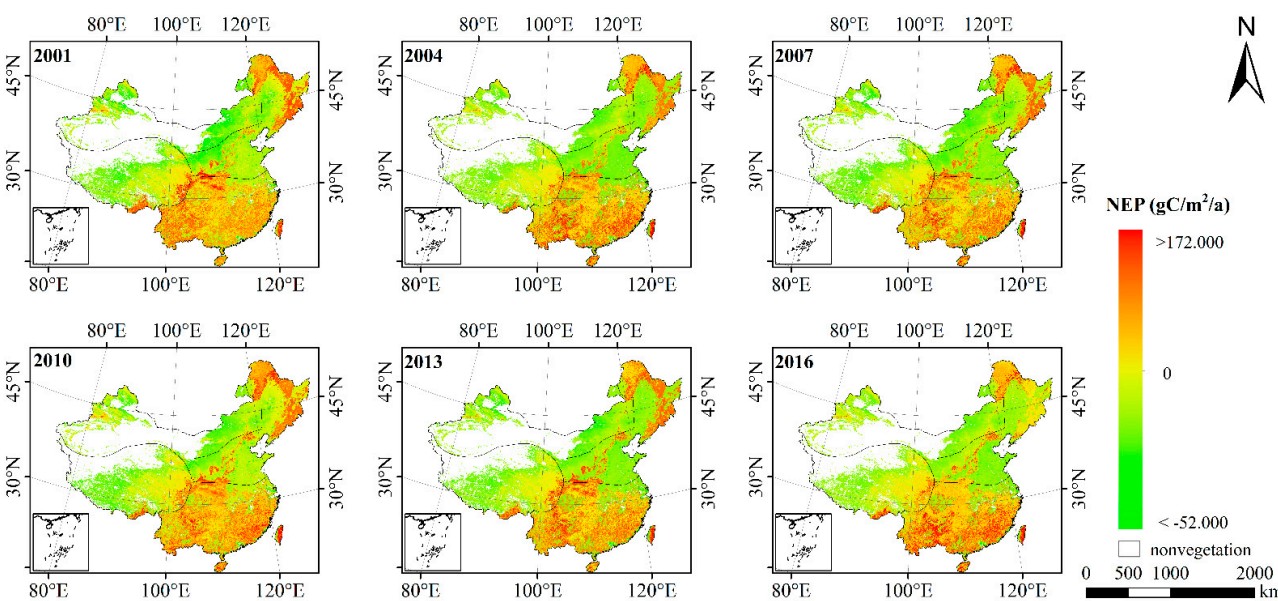

**Figure 8.** Average NEP in summer of China from 2001 to 2016.

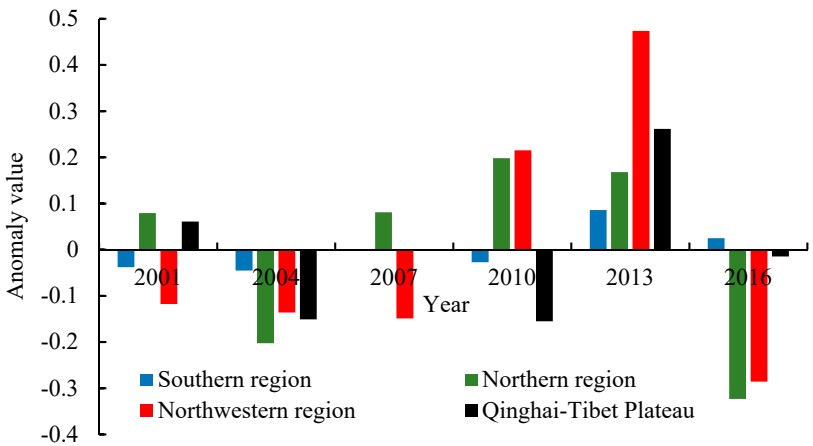

**Figure 9.** Inter-annual changes NEP in different geographic areas.

To analyze the temporal characteristics of NEP in different parts of China, the ANEP in different regions from 2001 to 2016 was analyzed (Figure 9). As can be seen from the figure, the ANEP in the four geographical regions from 2001 to 2016 can be divided into three stages. Specifically, before 2010, the NEP of each region was lower than the average level, indicating that Chinese carbon sink capacity was low at that stage. From 2010 to 2013, due to the progress of returning farmland to forest and greening in China, the NEP of each region was higher than the average level, showing an overall upward trend, indicating that Chinese carbon sink capacity was increasing. Then, from 2013 to 2016, the downward trend of Northwest China and Northern China was obvious.

In addition, it can be seen that the NEP of different geographical regions features spatial differences with time series. Southern China is a humid and rainy region, and vegetation grows well each summer. Therefore, the seasonal mean NEP values in this region are high and relatively stable in time series. Compared with the southern region, the NEP values in the northern region are relatively low, and fluctuations are relatively large in time series due to the influence of monsoon. Northwest China is an arid area with relatively sparse vegetation, so the absolute values of NEP are small. However, this region

is vulnerable to climate change, so the interannual variation of NEP is large. In the Qinghai Tibet Plateau, the main vegetation types are grassland and meadow, and the values of NEP in this region are relatively low and the interannual variation is also small.

- The trends of NEP from 2001 to 2016

To further explore the temporal and spatial change patterns of terrestrial ecosystem in China, according to Equation (22), the various trends of NEP in summer from 2001 to 2016 were calculated. Furthermore, the F test was used to determine the significance of changes. The data follow the F distribution, and the degrees of freedom are (1, n−2), where n represents 6 years. According to the F distribution table, $F_{0.05(1,4)} = 7.709$, $F_{0.01(1,4)} = 21.198$ According to these thresholds, the trend of the NPP and NEP can be divided into the following three levels: nonsignificant (F < 7.709), significant (F ≥ 7.709) and extremely significant (F ≥ 21.198). Moreover, as the *slope* values can be either positive (*slope* > 0) or negative (*slope* < 0), the trend of the NPP and NEP can be grouped into 6 levels: significant increase, significant decrease, slight increase, slight decrease, no significant increase and no significant decrease. The spatial variation trend was shown in Figure 10.

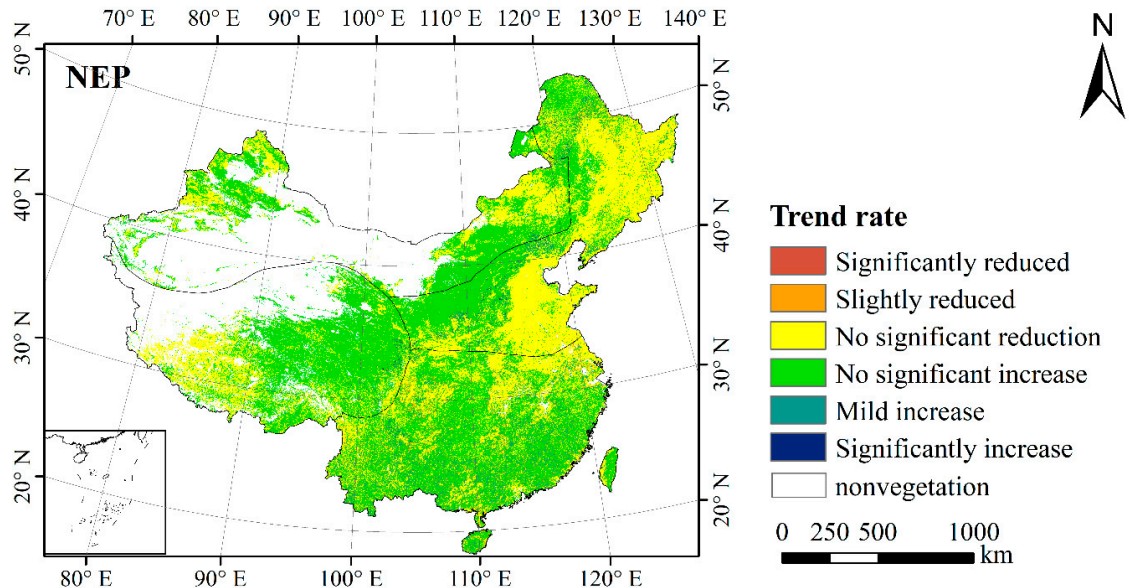

**Figure 10.** Trend rate of NEP in China during the summer from 2001 to 2016.

Figure 10 showed that the spatial variation trend of NPP and NEP during the summer of 2001 to 2016 was consistent, and except for the North China Plain, Northeast Plain, southwest Tibet, Northeast Inner Mongolia and some sporadic regions, the NPP and NEP in most regions of China showed an upward trend (albeit not reaching a significant level). Combining the figure of trend rate (Figure 10) and land cover types (Figure 1), it can be seen that the regions with a downward trend correspond to the main cropland of China, the grassland of Ngari in the Qinghai–Tibet Plateau, and the grassland of Xilin Gol League in Inner Mongolia. This suggests that the summer sink capacity of most cropland and some grassland in China is weakening.

There may be two reasons for the weakening of cropland carbon sink capacity. One is the change of soil respiration caused by agricultural fertilization and irrigation. Second, the non-agricultural process of cropland leads to the destruction of vegetation and release of carbon stored in soil (e.g., when the cropland is transformed into construction land, the plot becomes a carbon source). In addition, the decrease of NEP in the Ngari area of Qinghai–Tibet Plateau is caused by the overgrazing of grassland in this area [77]. As for the Xilin Gol League in Inner Mongolia, the main reason for the decrease of NEP is that the land desertification weakens the carbon sink capacity of vegetation [78].

## 4. Discussion

### 4.1. The NPP Estimation Results of Optimized CASA Model

The optimization of model parameters is very important to improve the NPP estimation accuracy of the CASA model. In the original CASA model, the parameter $\varepsilon_{max}$ of global vegetation was defined as 0.389 gC·MJ$^{-1}$ However, further research shows that it is difficult to obtain ideal results by using this value in different vegetation types [26,62,79,80]. This is because different types of vegetation have different $\varepsilon_{max}$ due to different photosynthetic capacity, leaf shape and canopy structure. Therefore, it is necessary to determine the value of $\varepsilon_{max}$ according to the vegetation type for model optimization. Based on the study of Zhu et al. [32,63,64], this study set $\varepsilon_{max}$ for CASA model according to vegetation cover data to estimate the NPP of terrestrial ecosystems in China. Meanwhile, the ratio of EET and PET was used to calculate $W_\varepsilon$ to reduce the model parameters and simplify the estimation process [31]. Compared with the original CASA model, the results have a higher accuracy, indicating that the optimized CASA model can be well applied to large-scale carbon source or sink estimation. In addition, referring to the measured NPP values [74], the estimated NPP of optimized CASA model basically falls within the range of the measured value, which indicates that the estimated result of the model also has a good reliability.

Based on the optimized CASA model, the average NPP of Chinese terrestrial ecosystem in 2010 was estimated to be 296.774 gC·m$^{-2}$·a$^{-1}$, which was consistent with the 273.500 gC·m$^{-2}$·a$^{-1}$ estimated by Li et al. [75]. In terms of vegetation types, Zhu et al. [74] and Sun et al. [81] showed that the NPP of EBF was the largest among all vegetation types, and their estimated NPP values for EBF were 1017 gC·m$^{-2}$·a$^{-1}$ and 972 gC·m$^{-2}$·a$^{-1}$, respectively. The NPP of EBF estimated in this study was 811.981 gC·m$^{-2}$·a$^{-1}$, which was also the highest of all vegetation types in this paper. In addition, the average NPP value of grassland calculated in this paper was 183.444 gC·m$^{-2}$·a$^{-1}$, which is consistent with the value of 194.260 gC·m$^{-2}$·a$^{-1}$ calculated by Liu et al. [82]. However, the NPP of ENF estimated in this study was slightly higher, which is different from the results of previous research. We think that the land use types of each study are different, which may lead to differences in NPP.

### 4.2. The NEP Estimation Results of Coupling Model

The current research mainly focuses on the estimation of NPP [81–84]. In fact, due to the existence of soil respiration, vegetation carbon sink needs to be characterized by m. The light energy utilization model including CASA model can only estimate NPP. Therefore, in order to obtain NEP, it is necessary to estimate soil heterotrophic respiration $R_h$. In this paper, the $R_h$ was estimated by GSMSR and $R_s$–$R_h$ relationship model. The analysis showed that when the NPP value was accurate, using $R_h$ obtained by the GSMSR and $R_s$–$R_h$ model could obtain NEP accurately ($R^2$ is up to 0.774). This result showed that it was feasible to couple the optimized CASA model with GSMSR and $R_s$–$R_h$ to realize large-scale vegetation carbon sink estimation.

The above coupling model was used to analyze the temporal and spatial changes of NEP in China. The results show that due to abundant precipitation and high temperature in the south, the NEP of terrestrial ecosystems in China is higher in the South and lower in the northwest and Qinghai Tibet. In addition, due to the difference in precipitation and temperature between winter and summer, the national NEP reached peaks in July and lowest value in January. The above results show that geographical and climatic conditions have an important impact on the spatial and temporal distribution of natural vegetation productivity, and the variation in climatic conditions (e.g., temperature and precipitation) may be the main reason for the change in vegetation productivity in most areas of China [85,86]. Further research will explore the correlation and influence mechanism between the climate conditions and NEP.

*4.3. The Prospects of the Study*

The results of this study show that the optimized CASA model can provide an effective way for NEP estimation. Nonetheless, the accurate estimation of NEP is also affected by the accuracy of $R_h$. The estimation of $R_h$ in this paper is based on geospatial statistical model and $R_s$–$R_h$ empirical model, which has certain uncertainties. Therefore, in order to obtain a more accurate NEP, it is necessary to optimize the measurement method of $R_h$ and further analyze the interaction between $R_h$ and NEP.

In addition, the summer NPP and NEP of terrestrial ecosystems in China from 2001 to 2016 are also studied. The results can enrich the study of carbon source/sink in China and provide a reference for China to create a policy on carbon balance. Nonetheless, due to the problems of data consistency and lack of spatial and temporal resolution, the spatial and temporal variation characteristics of Chinese NEP are further affected. Therefore, the production of remote sensing data products with higher spatial and temporal resolution is the key to achieve a more accurate spatial and temporal analysis of NEP.

## 5. Conclusions

NEP plays an important role in understanding ecosystem function and the global carbon cycle. In this paper, the NPP was estimated by using the optimized CASA model, and then the NEP was calculated by using GSMSR and the Rs–Rh relationship model. On this basis, the temporal and spatial variation patterns of NPP and NEP in China were analyzed, and the following conclusions were drawn:

(1) It is feasible to couple the CASA model with GSMSR and $R_s$–$R_h$ relationship model to estimate vegetation carbon sink, and model parameters optimization is an effective method to improve the estimation accuracy. Compared with the original CASA model, the $R^2$ of the optimized CASA model increased from 0.411 to 0.774, and the RMSE decreased from 21.425 $gC \cdot m^{-2} \cdot month^{-1}$ to 12.045 $gC \cdot m^{-2} \cdot month^{-1}$, indicating that it could improve the estimation accuracy by using vegetation classification to optimize the parameter $\varepsilon_{max}$ of the CASA model;

(2) Chinese NEP values are different in each region, presenting the pattern of southern region > northern region > Qinghai–Tibet region > northwestern region. From the annual average value of NEP, the southern and northern regions are carbon sinks as a whole, while the northwest and Qinghai Tibet regions are carbon sources. Nevertheless, the monthly variation patterns of NEP in different regions are generally similar, showing a single peak curve with summer as the peak;

(3) The NEP values of various vegetation types are also different. The annual average NEP values of vegetation types such as ENF, EBF, DBF and MXF are higher, and are presented as carbon sink; however, the NPP values of grassland and cropland are relatively lower and the $R_h$ values are relatively higher, so the mean NEP values are below zero, which shows that they are carbon sources. In addition, similar to different regions, the seasonal variation patterns of different vegetation also show a single peak curve with a peak in summer;

(4) The NEP in most regions of China show a non-significant level upward trend in the summer of 2001–2016, but main cropland and some grassland show a non-significant level downward trend. In addition, the NEP of different geographical regions have spatial differences with time series. The NEP in the south is much higher than that in the Qinghai–Tibet Plateau, but the fluctuations in the time series of both are relatively small; and that in Northern and Northwestern China are low, but their interannual changes are relatively large.

**Author Contributions:** Conceptualization, L.L. and D.G.; methodology, L.L. and D.G.; validation, D.G., L.L. and S.Q.; data curation, D.G. and L.L.; writing—original draft preparation, L.L. and D.G.; writing—review and editing, L.L., D.G., J.Y., S.Q., Y.S., S.W., L.W., L.Z. and J.K.; visualization, D.G. and L.L.; supervision, L.L.; project administration, L.L.; funding acquisition, L.L. and L.Z. All authors have read and agreed to the published version of the manuscript.

**Funding:** This research was supported by the Xuzhou Key R & D Project (KC19134), the Double carbon project of Jiangsu Normal University (JSNUSTZX202202), the National Natural Science Foundation of China (No. 41971305), the China Europe Dragon 5 Cooperation Programme (No. 59197), the Natural Science Foundation of Jiangsu Province (BK20181474), the Postgraduate Research & Practice Innovation Program of Jiangsu Province (KYCX20_2367, KYCX21_2625, KYCX21_1128), the Project Funded by the Priority Academic Program Development of Jiangsu Higher Education Institutions (PAPD).

**Data Availability Statement:** All freely available data are mentioned in section on Data and Methods.

**Acknowledgments:** The authors wish to thank for the data support from "National Geographic Resource Science SubCenter, National Earth System Science Data Center, National Science & Technology Infrastructure of China (http://gre.geodata.cn)".

**Conflicts of Interest:** The authors declare no conflict of interest.

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
