# Peer review of "Remote Sensing Estimation and Spatiotemporal Pattern Analysis of Terrestrial Net Ecosystem Productivity in China"

_remotesensing, doi:10.3390/rs14081902_

Round 1

Reviewer 1 Report

This manuscript conducts remote sensing estimation of NEP within China based on the CASA and GSMSR optimized models. In the process of NPP estimation, the introduction of complementary relationship reduces the required parameters, and the introduction of vegetation type data improves the accuracy of NPP estimation based on the CASA model. This study helps to understand the carbon source/sink in China. Although similar studies have already been carried out, this study is still an important supplement to China's carbon cycle research. In addition, the manuscript contains some grammatical errors and appropriate revisions are suggested. In general, I recommend reviewing the manuscript after major revision.

I also listed some of my comments as follows:

Major comments:

  1. Estimation of net productivity of terrestrial ecosystems is a very popular research direction, but as far as I know, the estimation of net productivity of terrestrial ecosystems and the analysis of temporal and spatial distribution of terrestrial ecosystems in China are relatively mature. For example: “Yao, Y., Li, Z., Wang, T., Chen, A., Wang, X., Du, M., ... & Piao, S. (2018). A new estimation of China's net ecosystem productivity based on eddy covariance measurements and a model tree ensemble approach. Agricultural and forest meteorology, 253, 84-93”. In other words, apart from using remote sensing estimation methods, what is new in your research compared with previous research, and has the estimation accuracy improved? This is critical for the application.
  2. Abstract needs to be reorganized. First, you should highlight the innovations of the research before introducing your methods. In addition, the introduction of the conclusion is too detailed, exceeding half of the content of the abstract, which is unreasonable, please describe the important conclusion.

Minor comments:

Line 18: “The NEP in a year was generally high in summer and low in winter”. This sentence seems to be common sense, the abstract should highlight more important findings, please delete it.

Line 34-37: There are some grammatical errors in this sentence, please modify the original sentence to “As China is one of the most diverse climates and ecosystems in the world, accurate estimates of NEP and analysis of spatiotemporal variation in NEP are critical to assessing the carbon balance of the Chinese terrestrial ecosystem”.

Line 44: It is recommended to unify “determination” as “estimation”, please unify in the full text.

Line 48: “over a wide range” is suggested to be changed to “at a regional scale”.

Line 57: The description should be specific to facilitate the reader's understanding, air temperature?

Line 58: Please remove “swath”.

Line 63: “factors” is suggested to be changed to “parameters”. In addition, such descriptions as "on the one hand.... on the other hand..." are rare in academic papers, please revise appropriately.

Line 86: Qinghai-Tibet also seems to belong to arid and semi-arid regions? Please explain it.

Line 100: "China's terrestrial ecosystem" is suggested to be revised to "Chinese terrestrial ecosystem".

Line 104: Chinese terrestrial ecosystem has both carbon source and carbon sink, so we should not only emphasize carbon sinks. Please change "carbon sink" to "carbon source and sink".

Line 127: The China Terrestrial Ecosystem Flux Research Network dataset seems to share 45 flux sites, why did the authors choose these eight sites? What is the specific reason?

Line 142-146: Is there a reference to the custom vegetation type? In addition, the MCD12Q1 land cover type product corresponds to one data per year. Has the dynamic change of vegetation been considered in the calculation process? This is very important.

Line 174: Please indicate the specific information of the software, such as version, release time, release location.

Line 182: What is the reference basis for selecting a SOCD with a depth of 20cm.

Line 309-310: The authors estimated NEP for 6 periods in 2001, 2004, 2007, 2010, 2013, and 2016, but only verified it for 2010, and it is questionable whether it is appropriate to do so.

Line 417-418: The full name has been mentioned in the previous text, and the abbreviation can be used in the following text. Please check the similar errors in the whole text.

Line 480 and 521: Polylines or patterns overlaid on text are not appropriate. please modify the figure.

Line 489: There is no period at the end of the sentence, please check the full text for similar errors.

Author Response

Major comments:

  1. Estimation of net productivity of terrestrial ecosystems is a very popular research direction, but as far as I know, the estimation of net productivity of terrestrial ecosystems and the analysis of temporal and spatial distribution of terrestrial ecosystems in China are relatively mature. For example: “Yao, Y., Li, Z., Wang, T., Chen, A., Wang, X., Du, M., ... & Piao, S. (2018). A new estimation of China's net ecosystem productivity based on eddy covariance measurements and a model tree ensemble approach. Agricultural and forest meteorology, 253, 84-93”. In other words, apart from using remote sensing estimation methods, what is new in your research compared with previous research, and has the estimation accuracy improved? This is critical for the application.

Reply: Thank you very much for your advice. The paper “A new estimation of China's net ecosystem productivity based on eddy covariance measurements and a model tree ensemble approach” is one of the important references cited in the manuscript (ref. 6). This paper and related research apply an MTE approach to upscale the flux NEE measurements to obtain NEP dataset. Its essence is to use the point source data of ground observation and combine other data to obtain spatially continuous data. In contrast, the manuscript uses remote sensing and meteorological data to estimate NPP based on the optimized CASA model, and then obtains NEP data set by coupling GSMSR and Rs-Rh relationship model. The method directly uses the spatial continuous data (remote sensing data) to realize the estimation of NEP. Since the original text is not clearly expressed, we have revised the introduction as required, and highlighted the innovation of optimizing model parameters such as  and .

Revised paragraph 1: “To solve the above problems, the monthly estimated evapotranspiration (EET) and monthly potential evapotranspiration (PET) were calculated by the regional EET model and Bouchet complementary relationship in this paper[29, 30]. We use the ratio of EET and PET to calculate  to reduce the model parameters and simplify the estimation process[31]. Moreover, based on the study by Zhu et al.[32] and International Geosphere Biosphere Programme (IGBP) classification data, we set optimized values for different vegetation types to improve the estimation accuracy of the CASA model.” (Please refer to the part marked in red in the revised version)

Revised paragraph 2: “In this paper, we will combine remote sensing data, meteorological data and soil organic carbon density (SOCD) data to estimate and analyze the NEP in China through the following steps: 1) optimize the parameter of CASA model by vegetation classification data to improve the estimation accuracy of NPP; 2) on this basis, use meteorological data and SOCD data to estimate the annual NEP and interannual NEP of Chinese terrestrial ecosystem by coupling the GSMSR and Rs-Rh relationship model; 3) use the estimated results to analyze the spatiotemporal distribution characteristics and change trends of NEP to provide support for carbon source and sink estimation and carbon cycle research in China.” (Please refer to the part marked in red in the revised version)

  1. Abstract needs to be reorganized. First, you should highlight the innovations of the research before introducing your methods. In addition, the introduction of the conclusion is too detailed, exceeding half of the content of the abstract, which is unreasonable, please describe the important conclusion.

Reply: Thank you very much for your valuable suggestions. We have rewritten the abstract as follows:

Abstract: Net ecosystem productivity (NEP) plays an important role in understanding ecosystem function and the global carbon cycle. In this paper, the key parameters of the Carnegie Ames Stanford Approach (CASA) model, maximum light use efficiency , was optimized by using vegetation classification data. Then the NEP was estimated by coupling the optimized CASA model, geostatistical model of soil respiration (GSMSR) and the soil respiration - soil heterotrophic respiration (Rs-Rh) relationship model. The ground observations from ChinaFLUX were used to verify the NEP estimation accuracy. The results showed that the R2 of the optimized CASA model increased from 0.411 to 0.774, and RMSE decreased from 21.425  to 12.045, indicating that optimizing CASA model by vegetation classification data was an effective method to improve the estimation accuracy of NEP. On this basis, the spatial and temporal distribution of NEP in China was analyzed. The research indicated that the monthly variation of NEP in China was a single peak curve with summer as the peak, which generally presented the pattern of southern region > northern region > Qinghai-Tibet region > northwest region. Furthermore, from 2001 to 2016, the most regions of China showed a non-significant level upward trend, but main cropland (e.g., North China Plain and Northeast Plain) and some grassland (e.g., Ngari in Qinghai-Tibet Plateau and the West Ujimqin Banner in Inner Mongolia) showed a non-significant level downward trend. The study can deepen the understanding of the distribution of carbon sources/sinks in China, and provide a reference for regional carbon cycle research. (Please refer to the part marked in red in the revised version)

Minor comments:

  1. Line 18: “The NEP in a year was generally high in summer and low in winter”. This sentence seems to be common sense, the abstract should highlight more important findings, please delete it.

Reply: Thank you for your suggestion. This sentence has been deleted and the whole abstract has been rewritten.

  1. Line 34-37: There are some grammatical errors in this sentence, please modify the original sentence to “As China is one of the most diverse climates and ecosystems in the world, accurate estimates of NEP and analysis of spatiotemporal variation in NEP are critical to assessing the carbon balance of the Chinese terrestrial ecosystem”.

Reply: Thank you very much for your attention to the details of the manuscript. This sentence has been revised according to your request. (Please refer to the part marked in red in the revised version)

  1. Line 44: It is recommended to unify “determination” as “estimation”, please unify in the full text.

Reply: Thank you for your suggestion. The whole manuscript has been revised as required.

  1. Line 48: “over a wide range” is suggested to be changed to “at a regional scale”.

Reply: Thank you for your suggestion. It has been modified according to your requirements.

  1. Line 57: The description should be specific to facilitate the reader's understanding, air temperature?

Reply: Thank you for your suggestion. It has been modified “temperature” to “air temperature”.

  1. Line 58: Please remove “swath”.

Reply: Thank you for your suggestion. The word “swath” has been deleted.

  1. Line 63: “factors” is suggested to be changed to “parameters”. In addition, such descriptions as "on the one hand.... on the other hand..." are rare in academic papers, please revise appropriately.

Reply: Thank you for your suggestion. The word “factors” has been modified to “parameters”, and the sentence has been modified as follows:

“First, the maximum light use efficiency  of global vegetation was defined as 0.389  in the original CASA model[21-23]. However, the value of  has always been controversial because it varies with different vegetation types[24-27]. In addition, the soil moisture submodel used to estimate the water stress coefficient () involves many physical parameters and it is difficult to obtain the data. The estimation results are affected by the spatial heterogeneity of soil[28].”

  1. Line 86: Qinghai-Tibet also seems to belong to arid and semi-arid regions? Please explain it.

Reply: Thank you for your question. I'm sorry for the misunderstanding caused by the unclear expression of the original text. This sentence should be:“for example, water is the main factor in the northwestern arid and semiarid areas, while temperature is the main factor in the Qinghai-Tibet and northeastern regions”. Northwestern arid and semiarid areas is a specific geographical region. (For details, please refer to the part marked in red in the revised version)

  1. Line 100: "China's terrestrial ecosystem" is suggested to be revised to "Chinese terrestrial ecosystem".

Reply: Thank you for your suggestion. It has been modified.

  1. Line 104: Chinese terrestrial ecosystem has both carbon source and carbon sink, so we should not only emphasize carbon sinks. Please change "carbon sink" to "carbon source and sink".

Reply: Thank you for your suggestion. It has been modified.

  1. Line 127: The China Terrestrial Ecosystem Flux Research Network dataset seems to share 45 flux sites, why did the authors choose these eight sites? What is the specific reason?

Reply: Thank you for your question. In line with the purpose that the verification data can be freely obtained for readers to verify, the verification data of this study are from the shared dataset of ChinaFLUX. At present, the shared dataset of ChinaFLUX only publishes the data of these eight sites (http://www.cnern.org.cn/data/initDRsearch?classcode=SYC_A02&tdsourcetag=s_pcqq_aiomsg). These eight flux site data sets are the ecosystem flux observation data of the first observation sites of ChinaFLUX, which can reflect the carbon and water fluxes of eight typical ecosystems including forests, grasslands and farmland in China(Wang, S. , Huang, K., Yan, H., Yan, H., Zhou, L., & Wang, H., et al. Improving the light use efficiency model for simulating terrestrial vegetation gross primary production by the inclusion of diffuse radiation across ecosystems in China. Ecological Complexity, 2015,23:1-13.  Ma, R., Zhang, L., Tian, X., Zhang, J., & Kato, T. Assimilation of remotely-sensed leaf area index into a dynamic vegetation model for gross primary productivity estimation. Remote Sensing, 2017, 9(3), 188.)

  1. Line 142-146: Is there a reference to the custom vegetation type? In addition, the MCD12Q1 land cover type product corresponds to one data per year. Has the dynamic change of vegetation been considered in the calculation process? This is very important.

Reply: Thank you for your suggestion. The original omitted references have been supplemented here (references 6, 19, and 55 in the revised manuscript). In addition, the dynamic changes of vegetation were considered in this study. Different classified data products are used in different years. I'm sorry that it was not clearly stated in the original manuscript, and it has been supplemented in the revised manuscript.

  1. Line 174: Please indicate the specific information of the software, such as version, release time, release location.

Reply: Thank you. It has been supplemented according to your suggestion. (For details, please refer to the part marked in red in the revised version)

  1. Line 182: What is the reference basis for selecting a SOCD with a depth of 20cm.

Reply: Thank you for your suggestion. The original omitted references have been supplemented (reference 59).

  1. Line 309-310: The authors estimated NEP for 6 periods in 2001, 2004, 2007, 2010, 2013, and 2016, but only verified it for 2010, and it is questionable whether it is appropriate to do so.

Reply: Thank you for your suggestion. As mentioned above, the validation data of this study are from the shared data of ChinaFLUX. At present, the data of ChinaFLUX is only up to 2010, and the data of 2010 is the most complete. Therefore, the current test only uses the data of 2010. After the data is updated, we will conduct further research. Thank you again for your advice.

  1. Line 417-418: The full name has been mentioned in the previous text, and the abbreviation can be used in the following text. Please check the similar errors in the whole text.

Reply: Thank you for your suggestion. It has been revised and checked throughout the manuscript.

  1. Line 480 and 521: Polylines or patterns overlaid on text are not appropriate. please modify the figure.

Reply: Thank you for your suggestion. It has been revised (Please refer to figures 7 and 9 of the revised manuscript).

  1. Line 489: There is no period at the end of the sentence, please check the full text for similar errors.

Reply: Thank you for your suggestion. It has been revised and checked throughout the manuscript.

Thank you again for your hard work for the manuscript!

Reviewer 2 Report

This manuscript has been studied the Terrestrial NEP of China through multi model coupling. In general, the manuscript has detailed data, correct methods, clear ideas and reliable conclusions, which is worthy of publication on Remote Sensing. Nevertheless, the manuscript still has the following problems that need to be modified:

  • In section 3.1.2, the author uses MOD17A3H product data to test the reliability of NPP data calculated in this manuscript. What are the advantages of NPP data estimated in this study compared with MOD17A3H products?
  • Line 423: “…left corner in Figure 3a exhibited…” According to the context, "figure 3a" should be changed to "figure 6a".
  • Line 459: “There were regional differences in the seasonal variation of NEP.” According to the context, “seasonal variation”should be changed to“monthly variation”.
  • Line 496: “…has more carbon sequestration and higher NEP……”. “carbon sequestration” should be unified as“carbon sink”.
  • Line 504:
  • Line 547-554: This part explains the reasons for the decrease of NEP in some parts of China. Some references can be cited appropriately so that readers who pay attention to the region can further understand the situation.
  • Section 4.1: The author's research shows that optimizing the maximum light energy utilization can effectively improve the accuracy of NPP estimation. In the discussion part, the author can briefly explore its internal causes from the perspective of plant physiology.

Author Response

  1. In section 3.1.2, the author uses MOD17A3H product data to test the reliability of NPP data calculated in this manuscript. What are the advantages of NPP data estimated in this study compared with MOD17A3H products?

Reply: Thank you for your question. MOD17A3H products only provide annual average data, while this study produces monthly average data products, which improves the time resolution. In addition, the data products in this manuscript can complement each other and cross verify.

  1. Line 423: “…left corner in Figure 3a exhibited…” According to the context, "figure 3a" should be changed to "figure 6a".

Reply: Thank you for your attention to the details. It has been corrected.

  1. Line 459: “There were regional differences in the seasonal variation of NEP.” According to the context, “seasonal variation”should be changed to“monthly variation”.

Reply: Thank you for your attention to the details. It has been corrected.

  1. Line 496: “…has more carbon sequestration and higher NEP……”. “carbon sequestration” should be unified as“carbon sink”.

Reply: Thank you for your suggestion. It has been modified.

  1. Line 504:Same as above

Reply: Thank you for your suggestion. It has been modified.

  1. Line 547-554: This part explains the reasons for the decrease of NEP in some parts of China. Some references can be cited appropriately so that readers who pay attention to the region can further understand the situation.

Reply: Thank you for your suggestion. The following references have been added to the paragraph:

  1. Duan, Y.; He, Z.; Wang Y.; Liu J.; Huang G. Monitoring land desertification of Tibet Autonomous Region based on remote sensing. Journal of Arid Land Resources and Environment. 2014, 28(1): 55-61.
  2. Zhao, Z.; Xu D.; Zhang X.; Lu Z.; Zhang X. Assessment of the Land Desertification Vulnerability in Inner Mongolia During the Period 2000-2015. Research of Soil and Water Conservation. 2020, 27(1): 168-175.

  1. Section 4.1: The author's research shows that optimizing the maximum light energy utilization can effectively improve the accuracy of NPP estimation. In the discussion part, the author can briefly explore its internal causes from the perspective of plant physiology.

Reply: Thank you for your suggestion. The following contents have been added to the manuscript as required:

“…However, further research shows that it is difficult to obtain ideal results by using this value in different vegetation types[26, 62, 79, 80]. This is because different types of vegetation have different  due to different photosynthetic capacity, leaf shape and canopy structure. Therefore, it is necessary to determine the value of  according to the vegetation type for model optimization …”

Thank you again for your hard work for the manuscript!

Round 2

Reviewer 1 Report

The paper has been revised according to the revision comments, and I agree to publish it.